# Counting atoms faster: policy-based nuclear magnetic resonance pulse sequencing for atomic abundance measurement

**Rohan Shenoy*** [1] [2]   **Evan A. Coleman*** [1]   **Hans Gaensbauer** [1]   **Elsa A. Olivetti** [1]

## Abstract

Quantifying the elemental composition of a material is a general scientific challenge with broad relevance to environmental sustainability. Existing techniques for the measurement of atomic abundances generally require laboratory conditions and expensive equipment. As a result, they cannot be deployed *in situ* without significant capital investment, limiting their proliferation. Measurement techniques based on nuclear magnetic resonance (NMR) hold promise in this setting due to their applicability across the periodic table, their non-destructive manipulation of samples, and their amenability to *in silico* optimization. In this work, we learn policies to modulate NMR pulses for rapid atomic abundance quantification. Our approach involves three inter-operating agents which (1) rapidly align nuclear spins for measurement, (2) quickly force relaxation to equilibrium, and (3) toggle control between agents (1) and (2) to minimize overall measurement time. To demonstrate this technique, we consider a specific use case of low-magnetic-field carbon-13 quantification for low-cost, portable analysis of foodstuffs and soils. We find significant performance improvements relative to traditional NMR pulse sequencing, and discuss limitations on the applicability of this approach.

## 1. Introduction

Developing scientific solutions to mitigate climate change can, in certain cases, be reduced to the task of budgeting or counting atoms of specific elements. These interpretations are commonly leveraged in multiple areas related to sustainability, including soil health measurement, material impurities management, and emissions monitoring (Norris et al., 2020). With such a simplification, this accounting process can be applied as an objective function for control-related tasks, which prevent those atoms from achieving wasteful or pollutant fates in industrial processes. The carbon credit marketplace is a macroscopic example, as it seeks to commodify carbon atoms by assigning a cost to their release (as greenhouse gases, or GHGs) and a reward to their sequestration (as e.g. carbonates, organic compounds, or pressurized gases) (Probst et al., 2024). However, such decarbonization strategies face a major challenge: accurately and verifiably monitoring these elemental transactions in a globally-scalable fashion (Coleman et al., 2023; Macfarlane et al., 2024). Many techniques are commonly employed to effectively count atoms (measure *atomic abundances*) through chemical reactions and other destructive processes under controlled conditions, but these techniques share fundamental limitations that hinder their effectiveness for climate objectives broadly.

These limitations motivate the development of a measurement alternative which can meet the needs of climate change solutions. In this work, we propose a scalable data-driven method for rapid atomic abundance measurement in challenging measurement environments. Our contribution is threefold. First, we train policies by reinforcement which shape and sequence magnetic field pulses for use in a simplified (low-field) nuclear magnetic resonance (NMR) spectroscope. Unlike incumbent procedures, this method is non-destructive, uses inexpensive and small-scale hardware, and generalizes to nuclear isotopes of many different elements. Secondly, we present a fast, robust simulator for generating large quantities of NMR spectroscopy data which is capable of reproducing the nuclear spin dynamics of many different samples in parallel when they are manipulated by an arbitrary magnetic field. Lastly, we present a novel method to manipulate these spins for atomic abundance measurement, by training three inter-operating agents to orchestrate an NMR pulse sequence. We demonstrate its performance on the task of carbon-13 ($^{13}$C) abundance measurement in caffeinated water, using a simulated dataset derived from laboratory NMR spectra, and find a significant performance improvement relative to incumbent techniques.

---

*Equal contribution   [1]Massachusetts Institute of Technology, Cambridge, MA [2]University of California Berkeley, Berkeley, CA . Correspondence to: Evan A. Coleman <ecol@mit.edu>.

*Proceedings of the $42^{nd}$ International Conference on Machine Learning*, Vancouver, Canada. PMLR 267, 2025. Copyright 2025 by the author(s).

## 2. Preliminaries

**Nuclear Magnetic Resonance (NMR)** is a physical phenomenon wherein the spins of atomic nuclei contained in a magnetic field are pulsed with electromagnetic radiation, causing them to precess before returning back in line with the magnetic field (Nishimura, 1996). This phenomenon is analogous to gently knocking a spinning top: it gyrates before eventually returning to stable rotation around the vertical axis. NMR spectroscopy is a measurement technique based on this phenomenon, in which a substance is placed in a strong background magnetic field and is exposed to a sequence of radio-frequency electromagnetic pulses which cause the atomic nuclei to radiate radio-frequency response signals at different frequencies according to their nuclear spin characteristics. The electromagnetic emission is measured at an axis perpendicular to the direction of the background magnetic field, giving a signal known as the free induction decay (FID). The Fourier transform of this signal is the NMR spectrum appropriate to the sample, and regression-based techniques are used to compare this spectrum to a library of reference spectra for identification and analysis. To evaluate the relative abundance of the characterized material in different sample, one generally compares the integrals of their spectra in a fixed window of frequencies. Many different pulse sequences have been designed to target various properties of atoms and molecular bonds. The most basic such pulses simply rotate spins of a given nuclear isotope by a fixed polar angle relative to the background magnetic field, with the extent normally denoted in radians as e.g. a $\frac{\pi}{2}$ or a $\pi$ pulse (Nishimura, 1996). NMR may be applied to solid-phase samples, but is typically and more readily applied to liquids (Reif et al., 2021).

**Magnetic Resonance Imaging (MRI).** NMR spectroscopy is commonly applied in medicine as part of a more sophisticated procedure known as MRI, which produces high-resolution anatomical imagery by associating FIDs to points in space, targeting the $^1$H nuclei in water (Nishimura, 1996). NMR and MRI both have many prevalent applications in food science, environmental monitoring, and various engineering fields (Marcone et al., 2013; Dais & Hatzakis, 2013; Soong et al., 2015; van der Klink & Brom, 2000).

**Machine Learning for Low-Resource MRI.** A current area of emerging NMR and MRI research is centered around developing and applying benchtop low-voltage or zero-field systems which use small permanent magnets or remove the background magnetic field entirely (Tayler et al., 2017; Zhao et al., 2024). These systems are of interest because of their reduced cost, as well as their ability to measure *in situ*. The approach thus represents a strong candidate for atomic abundance measurement, since it addresses many of the scalability concerns of laboratory-grade atomic abundance analyzers. Machine learning has been applied sparingly to

NMR in this context, but data-driven methods are commonly applied to MRI to reduce the time and resources required for medical imaging and diagnostics. Magnetic resonance fingerprinting is one such technique: it uses a pattern recognition algorithm on labeled data to derive a pulse sequence which can lead to direct imaging or segmentation of tissue properties (e.g. whether it is cancerous). The authors of (Ma et al., 2013) first introduced this method and demonstrated it on samples of brain tissue, finding a significant reduction in the required measurement time relative to standard approaches for the segmentation of grey matter from white matter (12.3 seconds versus nearly 10 minutes). More recently, deep learning was found to be highly effective at correcting electromagnetic interference in a low-field MRI setup (using a $0.05$ T magnetic field) without any loss in performance to a typical MRI (usually $\sim 2$ T) (Zhao et al., 2024). The authors showed that imaging of human subjects used less power and generated less noise with this approach, while requiring a comparable amount of measurement time.

**Reinforcement Learning (RL) for NMR.** Some recent work has applied reinforcement learning techniques to general molecular structure prediction with 80% accuracy (Sridharan et al., 2022), and to the classification of the geographic origins of wine with 94% accuracy (Korenika et al., 2024). However, neither of these works tuned the underlying NMR pulse sequence. Our proposed method is most similar to the earlier work in (Khaneja et al., 2005), which introduced gradient ascent pulse engineering (GRAPE). GRAPE is a form of model-predictive control (Schwenzer et al., 2021) which demonstrated that optimal NMR pulse sequences for coupled nuclear spin systems could be designed by gradient descent on simulated data to minimize distance to a general target state. However, the authors did not explicitly construct such a target state appropriate to abundance measurement.[1] In addition, GRAPE develops pulse sequences only *in silico*, and is only optimal given complete information about the intermediate states of the spins. It can become prohibitively computationally expensive for complex compounds containing large numbers of effective spins, which is an obstacle even in contexts where highly pure NMR spin systems are used, such as quantum computation (Jones, 2024). The output of GRAPE also requires an accurate

---

[1]**Note for domain experts:** To see why this is nontrivial, consider a spin-$\frac{1}{2}$ quantum system in the case of a 2D readout. Achieving a pulse which is agnostic to the direction of the target state in the transverse plane requires studying multiple pulses with different such targets. A reward which averages across such target states does not work, because it gives the direction of the background field $B_0$ as a target, i.e. $|C\rangle = \frac{1}{2\pi} \int_0^{2\pi} \left( |1\rangle + e^{i\theta} |0\rangle \right) = |1\rangle$. In this scenario, an optimal pulse which maximizes $\langle S_x^2 + S_y^2 \rangle$ is just $B_1 = 0$. Controlled experimental configurations to perform pulse-free measurement are challenging to engineer, as they require rapidly and repeatedly switching between measuring the sample and measuring pure noise, similar to optical chopper setups.

underlying physical model (Hamiltonian) for the spins in the sample, which is not generally possible for complex, long-chain molecules such as proteins (Hu et al., 2021).

**Our Contribution.** This work is the first to apply modern reinforcement learning approaches to NMR pulse sequencing for rapid atomic abundance measurement. Relative to methods such as GRAPE, we train policies by reinforcement on observables which are actually accessible in real NMR systems, so that our approach can exploit but does not rely on a complete underlying description. In addition, while we demonstrate the approach on simulated low-cost hardware, our method does not require an accurate pre-determined physical model of the underlying sample, and can be trained directly on real samples. We do not encounter computational barriers in our simulation-based results because we use the macroscopic Bloch equations to model the system, which are appropriate for a large sample of material (Nishimura, 1996). These are straightforward to simulate, and simplify the design of objectives.

### 2.1. Simulator Design

To train policies, we emulate nuclear spin dynamics for a general NMR spectrometer. Our NMR simulator is structured around an NMR environment that mimics a real-world spectrometer by restricting the agent's view to only high-level information that would be measured experimentally. Underneath this environment lies a spin simulator, which applies a first-order differential equation solver to the Bloch equations (see Appendix A), which model the time evolution of the individual nuclear spin vectors that compose a specified sample. Given an applied magnetic pulse, a set of nuclear spin parameters, and the chosen time resolution, the spin simulator computes the response of the sample and produces the net $M_x$ and $M_y$ magnetization components of the sample. Before this output vector is passed back to the NMR environment, Johnson-Nyquist instrument noise is introduced. The profile of this noise is roughly constant in frequency space (which we model as Gaussian in the time domain), and has an amplitude determined by the temperature of the readout, as well as its associated bandwidth and resistance (Perepelitsa, 2006).

The NMR environment receives the net calculated magnetizations and uses them to construct the state space presented to the agents. To demonstrate the approach, we assume the use of a two-axis magnetometer for signal readout, and a single-axis Faraday coil to generate the magnetic field for the pulse, though we emphasize that these are not requirements for the proposed method and can be easily changed based on available hardware. Each state is represented as a four-element tuple in two dimensions, comprising:

1. The observed transverse magnetization $(M_x, M_y)$

2. The newly applied pulse, $[B_1, 0, 0]$
3. The constant background $B_0$ field, $[0, 0, B_0]$
4. Instrument parameters: runtime T and temperature $\tau$.
5. The maximum observed magnetization observed $(\max_t \sqrt{M_{x_t}^2 + M_{y_t}^2})$ thus far.

The temperature (which controls the instrument noise level) evolves as $\tau_{t+1} = \tau_t * C$, where $C \sim N(1, \sigma_\tau)$, and C is redrawn at each timestep. While normally temperatures are held constant during runs, this choice of state space mimics the reduced capabilities of a low-cost *in situ* NMR spectrometer, and provides sufficiently realistic conditions for the agent to learn effective pulse modulation strategies.

We vectorize this environment using the `SubprocVecEnv` class in OpenAI Gymnasium, which runs multiple independent instances of the NMR in parallel. This considerably accelerates the training process by enabling effective training over modern multi-core hardware or on a distributed cluster. Furthermore, the agent is able to explore a diverse set of samples concurrently, mitigating overfitting to a single sample or specific simulator configurations. This setup also facilitates running multiple instances of the same sample in parallel, meaning that the that the noise inherent to each simulation instance is averaged over many environments, leading to more robust training and policies.

### 2.2. Laboratory Data and Spin Set Generation

Each atomic spin in our simulation is defined by a tuple $(\gamma, T_1, T_2)$, where $\gamma$ is the gyromagnetic ratio, $T_1$ is the longitudinal (spin-lattice) relaxation time, and $T_2$ is the transverse (spin-spin) relaxation time.[2] The gyromagnetic ratio $\gamma$ governs the rate at which a spin precesses around the static magnetic field $B_0$. The longitudinal relaxation time $T_1$ describes how quickly the the spin's precession speed decays after excitation, while the transverse relaxation time $T_2$ characterizes how rapidly spins lose phase coherence in the transverse plane, causing the observable signal to decay.

To build a realistic spin set, we begin with the FID signal produced by actual NMR measurements. We first dissolved pure caffeine powder (Millipore Sigma, 99% purity) in water at a concentration of 0.057 mol/L. We then performed a serial dilution of this sample and collected 6 FIDs at progressively lower caffeine concentrations. We used the default 1D NMR pulse (Single $\frac{\pi}{2}$, 8000 scans, 2s acquisition time per scan) in a JEOL502 two-channel ECZ spectrometer operating at 500.44 MHz. We apply a Fourier transform to these FIDs to obtain NMR spectra, which exhibits sharp peaks corresponding to different atomic spins $(\gamma, T_1, T_2)$

---

[2]**Note for domain experts:** As this paper is primarily addressed to a machine learning audience, for simplicity we did not introduce the concept of a chemical shift $\delta$ and instead label spins by individually-defined effective gyromagnetic ratios $\gamma$.

present in the sample. From the peak frequencies (also known as chemical shifts), we infer the gyromagnetic ratios $\gamma$ of constituent spins. We use the exponential decay of the FIDs to derive the average $T_2$ for the spins, and sample $T_1$s from a Gaussian distribution as a source of training data variability.

Next, we simulate a standard baseline one-dimensional NMR experiment using the identified gyromagnetic ratios. At this stage, the frequencies of the peaks in the simulated spectrum match those of the real samples. However, the intensities (peak heights) typically do not. We therefore adjust the weights associated with each spin so that the simulated peak heights align with those in the empirical spectrum. These weights roughly represent the relative numbers of each class of spin in the overall sample. By iterating this process of matching the simulated spectrum to the measured spectrum, we arrive at a spin set whose composition closely aligns with the true samples'.

## 3. Experiments

In pursuit of a framework for counting atoms with NMR, we explored and evaluated three reinforcement learning strategies which build on one another. The guiding idea behind our approach is to devise policies capable of manipulating nuclear spins to squeeze the largest response possible out of a given sample, under realistic environmental conditions. The "loudness" or intensity of this signal is expected to be directly proportional to the abundance of nuclear spins in a sample (Nishimura, 1996). **First**, we ask whether an agent trained to maximize this signal can perform better than a baseline NMR pulse. **Second**, we test whether a policy to quickly reset the system to its initial undisturbed state can permit this maximum response to be quickly reproduced. **Lastly**, we demonstrate that the decision to reset the system can also be delegated to an agent.

For all policy trainings in these experiments, we employed the Proximal Policy Optimization (PPO) algorithm (Schulman et al., 2017) provided in Stable-Baselines3 (Raffin et al., 2021). Additional details regarding this approach are provided in Appendix B.

### 3.1. Model 1: Making Samples Chirp by Maximizing Transverse Magnetization

In the first experiment, we aim to design a pulse which accurately and quickly determines the maximum achievable magnetization response for a compound of interest. Formally, we imagine placing a sample of material in a mild background magnetic field along the $z$ axis, and we reward an agent for achieving large values of $M_{x,t}^2 + M_{y,t}^2$, where $M_{x,t}$ and $M_{y,t}$ are the net observed magnetizations in the $x$ and $y$ directions (the directions of the magnetic field sensors

in the NMR), at timestep $t$. Intuitively, this phenomenon should occur when roughly all spins in the system have been coherently driven into the transverse plane, pointing in the same direction for a short period of time. This alignment causes a sharp and high-frequency jump in the observed magnetization, which we refer to as a *chirp*.

To formalize our approach, we define a Markov Decision Process (MDP), $(S, A, \Pi_{sa}, \Gamma, r)$, as follows:

- The state space, $S$, is defined above.
- The action space, $A$, is the continuous interval corresponding to $B_1 \in [-0.2, 0.2]$ Tesla, with $B_0 = 1$ Tesla. This magnetic field configuration is inexpensive to implement relative to those typically used in laboratory-grade NMR setups (generally ranging from 2 to 45 Tesla for $B_0$ and up to $\sim 0.1$ Tesla for Helmholtz coil-pulsed $B_1$) (Moser et al., 2017; Conradi, 2024). Importantly, unlike setups with stronger $B_0$, this choice of parameters can be achieved inexpensively using a Faraday coil[3] for $B_1$ and a neodymium-based permanent magnet for $B_0$ (Barón et al., 2023).
- The policy $\Pi_{sa}$ denotes the learned distribution of actions conditioned on the observed state.
- The discount factor $\Gamma$ is set to 0.99.
- The reward function at any timestep $T$ is given by:

$$r_T = \max_{0 \le t \le T} \left( M_{x,t}^2 + M_{y,t}^2 \right). \tag{1}$$

To evaluate the efficacy of this approach, we train the PPO model (as described above) for 5M timesteps of length 0.2 ms across 50 distinct environments with $\sigma_\tau = 0.01$. Each environment is initialized with a unique set of atomic spins determined by the procedure outlined in Section 2.1. We repeat model training on 5 fixed random seeds.

Figure 1 plots the maximum magnetization observed for a collection of samples versus the concentration of caffeine in those samples, comparing the learned pulse against the standard 1D NMR pulse. The depicted relationship for the learned policy is steeper and more monotone than that of the 1D NMR pulse, indicating that a more precise and direct relationship with concentration is achieved. The time required for these maximum concentrations to be reached was roughly 16% of the time required for a 1D NMR pulse sequence to complete, representing a significant reduction. However, because the policy drives the sample to a state of higher magnetization, it may require significantly more

---

[3] The field strength of a solenoid made of wound copper wire is $\mu NI/L$, with $\mu$ the magnetic permeability of air, $I$ the current flowing through the wire, and $N$ the number of coil turns in length $L$. For a 43-gauge enamel-wound copper wire of diameter $\sim 0.006$ cm, the required current for a singly-wound coil to achieve a $B_1$ of 0.2 Tesla is 9.7 Amperes, comparable to the current drawn by a home microwave.

time to relax to its initial state. This is a problem if the experiment is to be repeated many times in order to suppress measurement noise, suggesting the need for a second policy which can accelerate the sample's return to equilibrium. We address this concern in Experiment 3.2.

### 3.1.1. TRAINING WITH INCREASING NOISE

To understand the impact of increased noise on model efficacy, we modify the training procedure by progressively increasing the instrument noise profile applied to the observed magnetizations. The training begins with an insubstantial noise profile, allowing the model to initially focus on learning how its actions affect the spins without being hindered by noise. Every 500k timesteps, the noise level is multiplied by a factor of 10. This gradual increase in noise continues until the final 500k timesteps, at which point the noise profile is identical to that under which the model in the baseline Experiment 3.1 is trained. This staged approach introduces the agent to increasingly challenging conditions over time, while also preventing it from being overwhelmed by noise in the early stages. We find that this increased noise did not meaningfully change the capabilities of the learned policy at achieving a high magnetization, compared to the standard procedure used in Experiment 3.1. This indicates that the model is either inherently robust to noise variations or that the later stages of training, where noise levels match those in Experiment 3.1, dominate the learned policy. Consequently, our findings suggest that while progressive noise increases may prevent early instability, it does not meaningfully improve final performance.

### 3.1.2. DIRECT OPTIMIZATION OF STANDARD 1D NMR PULSE SEQUENCE

In this experiment, we initialize the agent with a pre-trained policy derived from from the standard 1D NMR pulse sequence. This approach leverages the structure and known efficacy of the standard pulse sequence as a prior, rather than starting from scratch. By doing so, we seek to understand whether features of existing approaches can be exploited to design effective pulse sequences by reinforcement. The process we apply here is analogous to the design of so-called "soft" NMR pulses, which are slight deformations of the standard or "hard" pulses (Brüschweiler et al., 1988).

To pre-train the model, we employ behavior cloning using the imitation learning framework from the Stable-Baselines3 library (Raffin et al., 2021). Observation-action sequences are generated by choosing the actions determined by applying the 1D NMR pulse sequence to the NMR environment, where the observations are determined by the state of the system, and the actions are the pulse values dictated by the standard pulse sequence. The model is trained for 50K epochs, with each epoch consisting of 2,048 timesteps in the

environment. During this phase, the agent learns to replicate the time-dependent behavior of the standard pulse sequence.

Once pre-training is complete, the neural network weights are transferred to the underlying network of the PPO agent. The agent is then trained for 5M timesteps in the environment, following the same procedure used in the baseline experiment detailed in Section 3.1. This approach allows the agent to refine the standard 1D NMR pulse using our reinforcement learning framework. Interestingly, the policy converged rapidly, within the first 300K timesteps, which suggests that the 1D NMR pulse may be a local minimum for the loss function.

This model proved to be the most effective among our three chirping models. As shown in Figure 1, it achieves a substantial increase in achieved magnetization, particularly in higher concentration samples. This suggests that leveraging the structured prior of the 1D NMR pulse sequence provides a strong initialization, allowing the model to refine its policy more efficiently and exploit domain-specific patterns that improve performance.

### 3.2. Model 2: Accelerated Spoiling

The standard 1D NMR pulse sequence alternates between applying a sinusoidal pulse and allowing the system to reset (i.e. not pulsing at all) during a fixed period. This ensures that the system resets to the equilibrium state, so that repeated pulses (or scans) can generate statistically-independent samples of the same signal. The effect of repeated measurement is to suppress noise at a rate proportional to $\frac{1}{\sqrt{n}}$, where $n$ is the number of scans. While this conservative approach ensures accuracy, it is time-intensive. An approach known as *spoiling* can be employed to more quickly nullify residual transverse magnetization (Jehenson & Bloch, 1991).

In this experiment, we learn an accelerated spoiling policy. The MDP is the same as in Experiment 3.1, but the reward at any timestep $t$ is modified to

$$r_t = -(M_{x,t}^2 + M_{y,t}^2) - \mathbb{1}[t = T](M_{x,T}^2 + M_{y,T}^2). \quad (2)$$

This reward penalizes the model for the current transverse magnetization at any step $t$, as well as the final transverse magnetization at the end of the episode.

We find that the learned spoiling policy is highly effective at rapidly reducing transverse magnetization by actively dispersing the spins from the transverse plane, which is how the agent receives the system state after the chirp. Compared to the standard method of simply applying no pulse, our approach achieves an up to 28% reduction in average magnetization over the evaluation period. This effect can be visualized in Figure 2, where the policy outperforms passive relaxation in driving the system to lower magnetization.

Importantly, this accelerated relaxation occurs within a timescale that is just a fraction of the system's $T_2$ relaxation time, whereas in the standard protocol, the system is allowed to relax for multiple $T_2$ periods. While the system is not reset to equilibrium within this shortened relaxation period, the suppression of magnetization is significant enough that we can simply terminate the pulse after a short spoiling time $t_S$ and achieve equilibrium more rapidly than the incumbent approach.

### 3.3. Model 3: Toggled Chirping and Spoiling

In standard NMR spectroscopy, the spoiling period must reset the transverse magnetization to below a threshold, so that the sample is returned to its initial state before repeating the pulse. In this experiment, we consider the possibility of instead permitting a partial reset, by training a third policy to choose which of the chirping or spoiling models has control of the environment. After chirping with the model trained in Section 3.1.2, the spoiling model from Section 3.2 can take over and push the spins toward equilibrium, before the next chirp re-establishes high magnetization. The MDP appropriate to this process has the same state space, but with an additional binary flag indicating which of the chirp or spoiling policies can control $B_1$, as well as the time $t_C$ since the chirping policy last had control and the time $t_S$ since the spoiling model last had control. The action space is a change to that binary flag, although this action is only available 200 timesteps after the most recent toggle. The reward function at any given timestep $T$ is taken as the change in maximum magnetization, with a penalty for long chirp-spoil sequences:

$$r_T = t_S \times \Delta\Big[\max ||\vec{M_t}||^2\Big]\Big|_{t<T-t_C}^{t<T} - t_C \times ||\vec{M_t}||^2 \quad (3)$$

In Figure 4, we demonstrate that our toggle model outperforms the chirp-only model due to the repeated partial reset facilitated by the spoiling stage. This result highlights the importance of the interplay between the chirp and spoiling models, which together enable us to maximize magnetization, quickly diminish transverse magnetization, and subsequently restore high magnetization in a cyclic manner.

To further validate the necessity of the spoiling model within this approach, we compare our toggle models performance against a modified chirp procedure where the maximum magnetization is artificially reset at the same points as the toggling transitions. This experiment reveals that simply setting the maximum magnetization to zero while the chirp is running is insufficient for maximizing magnetization over many runs. This limitation indicates that our MDP structure effectively learns to associate states with their maximum magnetization history, meaning that the learned policy partially depends on knowledge of prior magnetizations. Thus,

we demonstrate that the benefit of the spoiling model within this procedure arises solely from its ability to rapidly reduce the transverse magnetization.

To better understand the behavior of the chirp and spoiling models within the toggle framework, we visualize the models' actions as a function of the observed magnetizations as well as the corresponding state densities in Figure 3. Across different environments and samples, we observe that the chirping model consistently reaches the maximum magnetization in the same direction, validating the structured control it asserts in the system. Additionally, we find that the spoiling model applies a large RF pulse in a consistent manner until the magnetization is significantly reduced, at which point it will occasionally stop pulsing. The effect of this RF pulse is evident in the state density shift in the bottom two plots, where the distribution of magnetization states rapidly transitions in the direction of the RF pulse applied by the spoiling model (the x direction). These findings highlight the complementary roles of the chirping and spoiling models within our toggling framework. Crucially, we demonstrate that toggle between these models provides a superior strategy relative to simple relaxation for maintaining high magnetization and improving scan speed.

## 4. Limitations and Future Work

In this work, we introduced a novel technique to measure atomic abundances using NMR pulse sequences which are learned by reinforcement. While we demonstrate the success of our approach on simulated samples of caffeine dissolved in water, it has fundamental limitations which constrain its applicability and require discussion. Most crucially, the use of this method makes the implicit assumption that the samples being measured with a given pre-trained chirping policy have same relative distributions of nuclei as the samples used for training the pulse. As a result, any such application of this measurement approach in the field will require periodic cross-checks against traditional NMR to ensure that distribution shift from the training data has not occurred. Moreover, in this work we treated NMR pulses as targeting individual atomic spins, but inter-nuclear interactions within a sample such as proton coupling (Nishimura, 1996) can cause other elements to be activated by the applied magnetic field, leading to radio-frequency responses which interfere with our proposed procedure. Characterizing the performance of this proposed method as applied on real samples of analytes of practical interest, in low-cost hardware, is the clearest and most intriguing avenue of future research. We note in addition that the results in Figure 3 (top left, black dots) suggest the existence of a preferred direction for abundance measurement in the transverse plane. It would be of interest to compare our method against a variant of GRAPE which uses this vector as a target state.

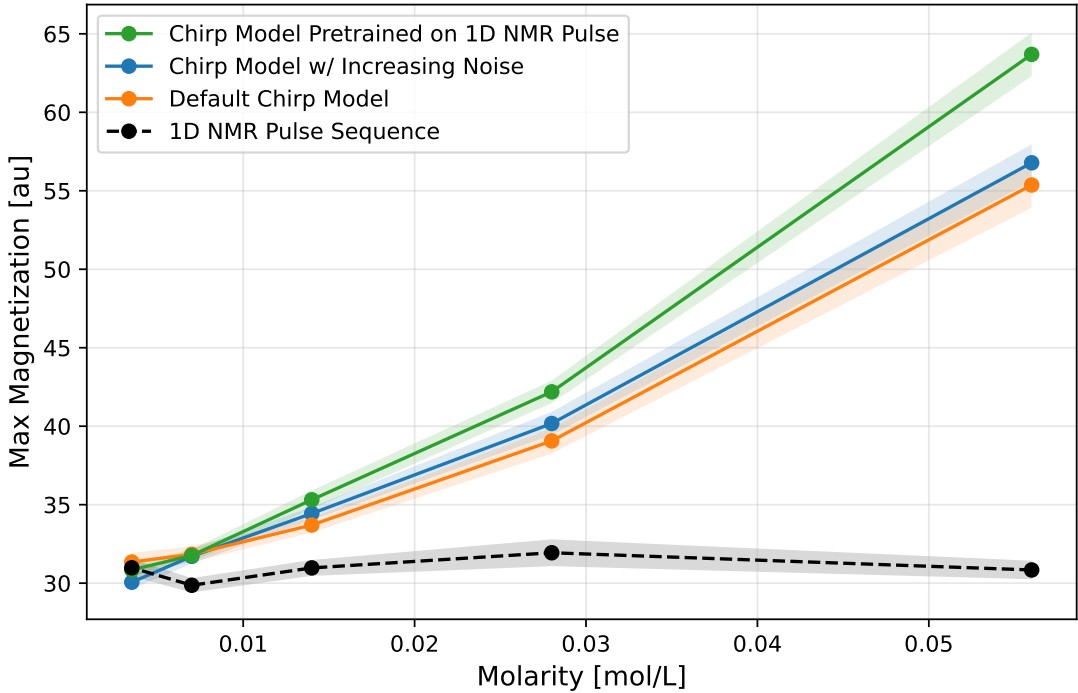

*Figure 1.* The maximum magnetization achieved by our three chirping models, with a baseline of the 1D NMR pulse sequence. The mean and standard error is across the five different fixed training seeds, as well as five environments (which can be interpreted as five independent scans). We see a clear one-to-one relationship between molarity (concentration) and maximum magnetization for our trained chirping policy. This was not the case for the incumbent 1D NMR approach (black line).

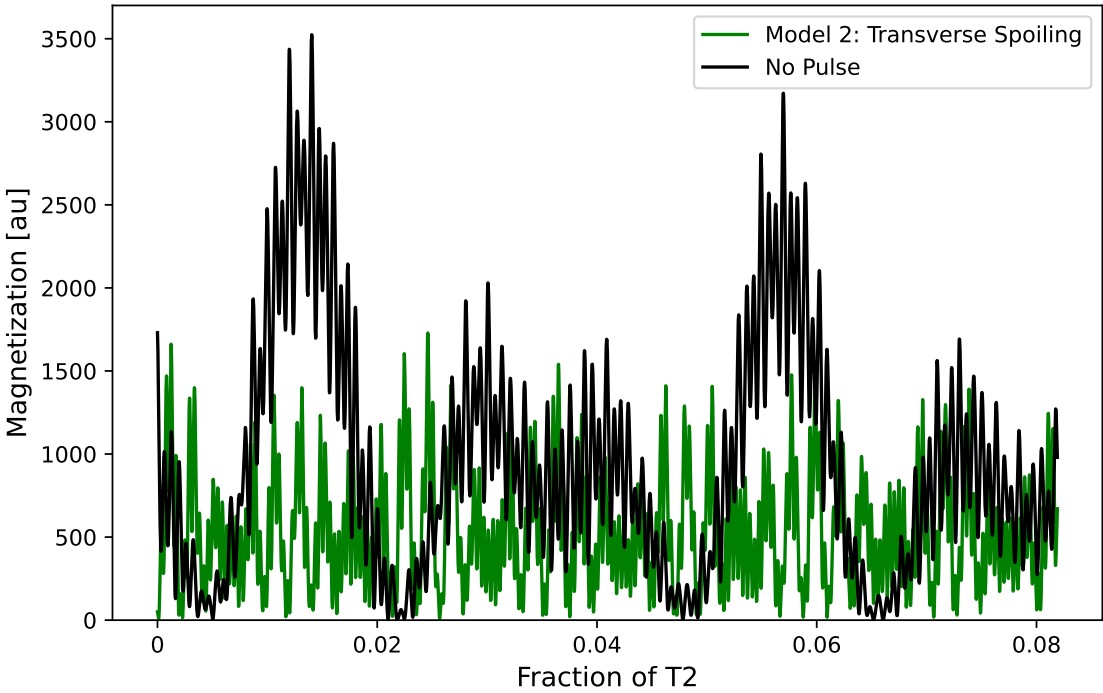

*Figure 2.* Demonstration of the performance of our spoiling model in reducing the average magnetization in one environment compared to the standard approach of applying no pulse.

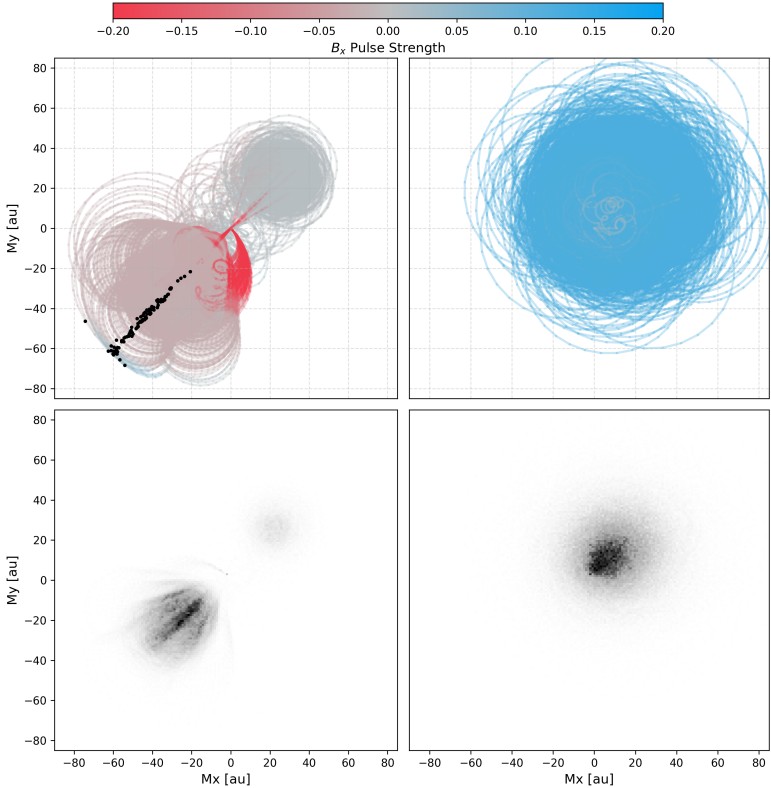

*Figure 3.* Heatmaps of the state-action relationship averaged over 50 independent scans of a sample. Lines represent the observed state of the system, components of the transverse magnetization. Lines are weighted by (Top) the action of the model, and (Bottom) time. Plots on the left visualize the chirping model's behavior, while those on the right show the spoiling model's behavior. In the top left, we include black dots to indicate where the max magnetization was achieved in each environment.

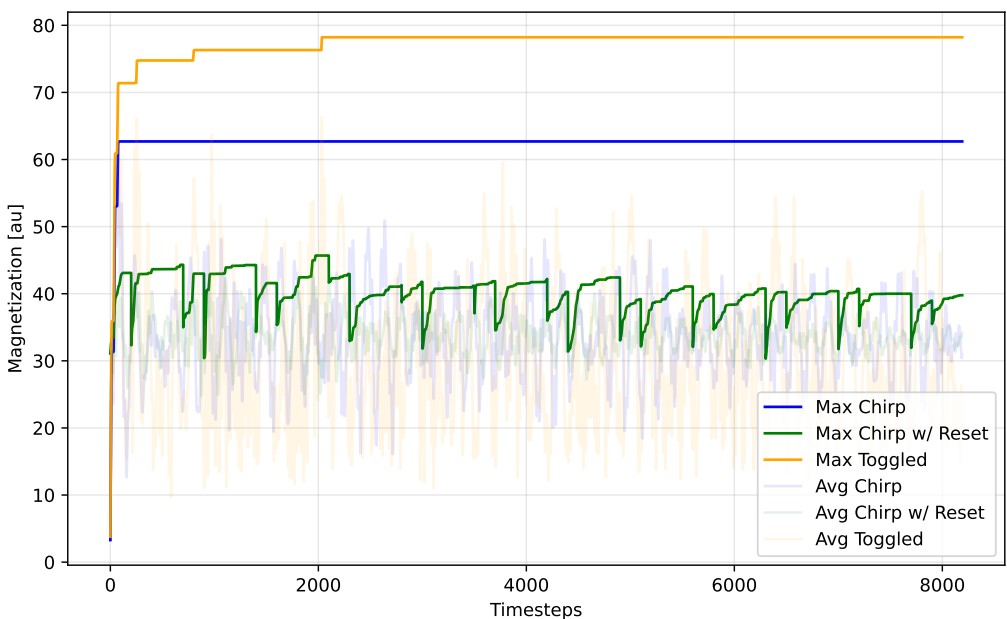

*Figure 4.* Depiction of the maximum and average magnetizations across three different approaches: (1) Chirping model's control; (2) Chirping model's control, but with the tracked max magnetization reset to $0$ whenever the toggle model acts; and (3) Toggle model's control.

## Acknowledgments

This work was partially funded by the MIT Climate and Sustainability Consortium. The authors are grateful for the feedback of attendees of the Climate Change AI workshop at NeurIPS 2024, "Tackling Climate Change with Machine Learning." The authors are additionally grateful to Glen Junor for support in the data collection process.

## Impact Statement

This work presents a novel reinforcement learning-based approach to optimize nuclear magnetic resonance (NMR) spectroscopy for the purpose of atomic abundance measurement. We demonstrate that utilizing reinforcement learning to optimize the pulse sequence of these NMR systems can provide accurate measurements while reducing acquisition time. This approach has applications in environmental monitoring, materials science, and chemical analysis by making NMR-based sample property measurements more rapid, and allowing measurement to occur *in-situ* (Vyalikh et al., 2024; Simpson et al., 2018).

There are several potential implications of this work. First, the configuration of our simulator as a low-field NMR spectroscope means that these techniques can be applied to systems costing several hundred times less than a traditional NMR spectroscope (Michal, 2020). Second, the demonstrated efficacy of our methods within the broader NMR framework may build interest in machine learning-based pulse sequencing, revisiting and advancing prior discoveries in the field which relied on model-predictive control (Khaneja et al., 2005). Nontrivial applications of interest for our setup include rapidly quantifying the water-soluble carbon content of soils, which could have significant implications for nature-based carbon crediting systems, and enable more accurate and scalable verification of soil carbon sequestration. Due to its modularity, in agricultural settings this approach can additionally support data-driven optimization of fertilizer application and other precision farming techniques, improving food system sustainability and reducing its resource consumption.

While this work demonstrates promising results, several limitations must be considered. First, all experiments were conducted in simulation using a simplified dataset, meaning that real-world implementation will likely present additional challenges. Deploying this approach in a production NMR system would require significant computational resources and energy consumption to train and deploy, i.e. it has an environmental footprint which should be transparently documented. Additionally, the accuracy of this method depends on training the policy over a sufficiently diverse distribution of samples. If the model encounters out-of-distribution samples, its measurements could become

unreliable, posing potential risks in applications such as carbon crediting and precision farming. Inaccurate carbon sequestration estimates could lead to financial misallocation in carbon markets, while errors in soil composition analysis might result in improper fertilizer application, negatively impacting both crop yields and environmental sustainability. Ensuring robustness across a broad range of sample variations will therefore be critical for the practical deployment of this method.

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

## A. Bloch equations

The Bloch equations describe the time evolution of magnetization observed in an NMR system. The relevant variables are:

- $M_x, M_y, M_z$, the components of the magnetization vector $\vec{M}$.

- $B_x, B_y, B_z$, the components of the external magnetic field $\vec{B}$.

- $\gamma$, the nuclear spin's gyromagnetic ratio.

- $T_1$, the longitudinal (spin-lattice) relaxation time.

- $T_2$, the transverse (spin-spin) relaxation time.

- $M_0$, the equilibrium magnetization.

The corresponding system of coupled differential equations is:

$$\frac{dM_x}{dt} = \gamma(M_y B_z - M_z B_y) - \frac{M_x}{T_2}, \tag{4}$$

$$\frac{dM_y}{dt} = \gamma(M_z B_x - M_x B_z) - \frac{M_y}{T_2}, \tag{5}$$

$$\frac{dM_z}{dt} = \gamma(M_x B_y - M_y B_x) - \frac{M_z - M_0}{T_1}. \tag{6}$$

## B. Policy architecture and supporting information

Here, we outline the structuring of the Markov Decision Process (MDP) and the architecture of the Proximal Policy Optimization (PPO) agent as implemented in Stable-Baselines3 (Raffin et al., 2021). The parameters of the policy network and the training pipeline are contained in Algorithms 1 and 2.

In this work, we structured our problem statement to employ a Markov Decision Process (MDP), a standard formalism for decisionmaking under uncertainty which is frequently applied in the context of reinforcement learning. An MDP is defined by the tuple $(S, A, \Pi_{sa}, r, \Gamma)$, with:

- $S$: the set of possible *states* the system can occupy.
- $A$: the set of possible *actions* the agent can take. The system $(S, A)$ is termed the *environment*, and it evolves with *transition probabilities* $\mathbb{P}(s|s', a)$, which define the probability of that the environment evolves to state $s$ after action $a$ is taken in state $s'$. In this paper, the transition probabilities effectively reduce to delta functions when we disable Johnson-Nyquist noise, because the Bloch equations are deterministic.
- $\Pi_{sa}$: the *policy* which determines the agent's behavior to manipulate its environment. While deterministic policies are possible, in our case the policy is randomly distributed and modeled with parameters $\theta$ as $\Pi_{sa} = \mathbb{P}(a|s) \sim \pi_\theta(a|s)$. [4]
- $r(s, a)$: the *reward function* which maps each state-action pair to a scalar signal indicating the desirability of the state with respect to a specified metric. The agent's policy is trained to maximize this reward. Note that the reward functions used in this work implicitly depend on $a$ but are specified only as functions of the state, $r(s)$.
- $\Gamma \in [0, 1)$: the *discount factor* which is responsible for balancing immediate rewards against future rewards. It multiplies contributions to the cumulative reward in each timestep, controlling the divergence of cumulative rewards over time and enabling the agent to drive the environment toward optimal outcomes on a long time horizon.

In many real-world settings the agent may not observe the full environment, but instead a state $s_t$ which is a subset of all available observables. This was the case for our setup, and we explain which values are masked in the details to follow.

---

[4] We avoided the more standard notation $\Pi_{sa} = \pi$ in the main body of the paper so as to avoid confusion with the angular shifts induced by 1D NMR pulses, e.g. "pulsing with $\pi$" versus "pulsing by $\pi$."

At each timestep, the agent observes the current state $s_t$, chooses an action $a_t \sim \pi_\theta(\cdot|s_t)$ according to its policy, receives a reward $r_t = r(s_t, a_t)$, and transitions to a new state $s_{t+1} \sim \mathbb{P}(\cdot|s_t, a_t)$. The goal of the agent is to learn parameter values that maximize the expected return, defined as the discounted sum of rewards:

$$\mathbb{E}_\pi \left[ \sum_{t=0}^{\infty} \Gamma^t r_t \right]. \tag{7}$$

The structure provided by an MDP allows for the development of gradient-based policy optimization methods such as the Proximal Policy Optimization (PPO) agent developed in this work. The agent's behavior is iteratively improved by updating a control policy which samples its actions probabilistically. The policy is learned in a data-driven fashion through interactions with the environment (of which it observes a fraction, the state-space $S$) and estimating updates. As reward function will generally be noisy or sparse, it can be difficult to estimate the payoff attributable to a given state. In the PPO agent we use in this analysis, we model it by a *value function* $V_\phi$, which has learnable parameters $\phi$. To guide learning, it estimates the expected future return $R$ starting from state $s$. Formally,

$$V_\phi(s) \approx \mathbb{E}_\pi \left[ \sum_{t=0}^{\infty} \Gamma^t r_t \, \middle| \, s_0 = s \right]. \tag{8}$$

The parameters $\phi$ are learned by minimizing the squared difference between the predicted value and the observed return:

$$L_{\text{value}} = \frac{1}{M} \sum_t \left( V_\phi(s_t) - \hat{R}_t \right)^2, \tag{9}$$

where the estimated return $\hat{R}_t$ is defined as the value of the state at time $t$ plus some estimate of advantage $\hat{A}_t$:

$$\hat{R}_t = V_\phi(s_t) + \hat{A}_t. \tag{10}$$

In our PPO implementation, the advantage $\hat{A}_t$ is estimated from temporal differences $\delta_t$ using Generalized Advantage Estimation (GAE) (Schulman et al., 2018). This means that after collecting experience, we compute the following recursively, using knowledge of future states:

$$\delta_t = r_t + \Gamma V_\phi(s_{t+1}) - V_\phi(s_t) \tag{11}$$

$$\hat{A}_t = \delta_t + \Gamma \lambda \hat{A}_{t+1} \tag{12}$$

where $\lambda$ is a tunable parameter. These definitions enable the basic training pipeline we outline in Algorithm 2, which learns $\phi$ and $\theta$ as defined in Algorithm 1.

We list below the major similarities of the Stable-Baselines3 PPO implementation with respect to the original work in (Schulman et al., 2017).

- **On-policy.** Policy is updated based on the behavior of the very policy being trained, not on the behaviors of other policies. Samples are not reused as in off-policy approaches.

- **Clipping to constrain updates.** Gradient clipping for actor by default, optional gradient clipping for value function (disabled by default). No trust region, no use of second-order derivatives or constraints to tune or stabilize.

- **Advantage estimation.** By default, uses Generalized Advantage Estimation (GAE) (Schulman et al., 2018).

- **Actor-critic.** Policy network $\pi_\theta$ is distinct and separate from value network $V_\phi$.

- **Environment handling.** Adapts to vectorized environments and allows for both continuous and discrete action spaces.

We emphasize that our policy network is a simple dense network, see Algorithm 1. We did not consider timeseries variants of this architecture which apply LSTMs or similar recurrent approaches to condition model actions on the history of the state. We also note that the Bloch equations (written in Appendix A) are local in time $t$, so that complete information about the state at any given time is sufficient to infer the next timestep (i.e. the dynamics are Hamiltonian). Within those equations, we effectively mask the $M_z, M_0, T_1, T_2$, and $\gamma$ values from the agent, so a non-recurrent architecture is not guaranteed to perfectly reconstruct or manipulate these dynamics. Nevertheless, it is interesting that our approach is performant as-is without such architectural considerations. Future work will investigate whether recurrent approaches improve the performance above this baseline.

---

**Algorithm 1** Default PPO MlpPolicy Architecture in Stable-Baselines3

---

1: **Input:** Observation $s \in \mathbb{R}^7$ ($\mathbb{R}^{10}$ in the case of the toggle policy, to account for $t_C$, $t_S$, and the binary flag)

2: **Actor network for magnetic field:**
   $\mu_\theta(s) \leftarrow \text{MLP}_{\text{actor}}(7 \text{ or } 10, [64, 64], 1)$       $\triangleright$ MLP with 2 hidden layers of size 64 and tanh activations
   $\log \sigma_\theta$ is a trainable scalar, shared across actions
   $\pi_\theta(a|s) = \mathcal{N}(\mu_\theta(s), \sigma_\theta^2)$

3: **Actor network for binary toggle flag:**
   $f_\theta(s) \leftarrow \text{MLP}_{\text{actor}}(10, [64, 64], 1)$       $\triangleright$ MLP with 2 hidden layers of size 64 and tanh activations
   $\pi_\theta(a|s) = \text{Bernoulli}(f_\theta(s))$
   Equivalently, $\pi_\theta(a = 1|s) = \text{sigmoid}(f_\theta(s))$

4: **Critic Network:**
   $V_\phi(s) \leftarrow \text{MLP}_{\text{critic}}(7 \text{ or } 10, [64, 64], 1)$       $\triangleright$ MLP with 2 hidden layers of size 64 and tanh activations

5: **Initialization:** Orthogonal
   - Gain $\sqrt{2}$ for all hidden layers
   - Gain 0.01 for actor output
   - Gain 1.0 for critic output

6: **Stochastic action sampling:** $a \sim \pi_\theta(a|s)$, $a \in \mathbb{R}^1$

---

---

**Algorithm 2** PPO training pipeline in Stable-Baselines3, including hyperparameters used to produce the final results.

---

**Require:** Initialized policy $\pi_\theta$ and value function $V_\phi$.

1: **for** each update iteration **do**
2:     **for** each timestep $t = 0, \ldots, T$ **do**                               $\triangleright$ Collect experience over $T = n_{\text{steps}} \times n_{\text{envs}}$ transitions
3:         Observe state $s_t$, take action $a_t \sim \pi_\theta(\cdot|s_t)$
4:         Record $(s_t, a_t, r_t, \log \pi_\theta(a_t|s_t), V_\phi(s_t))$
5:     **end for**

6:     $\hat{A}_T \leftarrow 0$
7:     **for** each timestep $t = T-1, \ldots, 0$ **do**           $\triangleright$ Estimate advantages $\hat{A}_t$ using GAE (Schulman et al., 2018)
8:         $\delta_t \leftarrow r_t + \Gamma V_\phi(s_{t+1}) - V_\phi(s_t)$,     $(\Gamma = 0.99)$
9:         $\hat{A}_t \leftarrow \delta_t + \Gamma\lambda\hat{A}_{t+1}$            $(\lambda = 0.95)$
10:       $\hat{R}_t \leftarrow \hat{A}_t + V_\phi(s_t)$                                        $\triangleright$ Compute returns
11:     **end for**

12:     **if** advantage normalization enabled (True) **then** $\hat{A}_t \leftarrow \frac{\hat{A}_t - \text{mean}(\hat{A}_t)}{\text{std}(\hat{A}_t) + 10^{-8}}$
13:     **end if**

14:     **for** $K$ epochs **do**
15:         **for** each minibatch of size $M = 64$ **do**
16:             $\tilde{r}_t(\theta) \leftarrow \frac{\pi_\theta(a_t|s_t)}{\pi_{\theta_{\text{old}}}(a_t|s_t)}$                         $\triangleright$ Compute importance sampling ratio
17:             $L_t^{\text{CLIP}}(\theta) \leftarrow \min\left(\tilde{r}_t(\theta)\hat{A}_t, \text{clip}(\tilde{r}_t(\theta), 1-\epsilon, 1+\epsilon)\hat{A}_t\right)$,     $(\epsilon = 0.2)$       $\triangleright$ Clip batch loss
18:             $L_{\text{policy}} = -\frac{1}{M}\sum_t L_t^{\text{CLIP}}$                               $\triangleright$ Compute policy loss

19:             **if** value clipping enabled (False) **then** $V_{\text{pred}} \leftarrow V_{\text{old}} + \text{clip}(V_\phi(s_t) - V_{\text{old}}, -\epsilon_{\text{vf}}, \epsilon_{\text{vf}})$
20:             **else** $V_{\text{pred}} \leftarrow V_\phi(s_t)$
21:             **end if**

22:             $L_{\text{value}} = \frac{1}{M}\sum_t \left(V_{\text{pred}} - \hat{R}_t\right)^2$                        $\triangleright$ Compute value loss
23:             $L_{\text{entropy}} \leftarrow -\frac{1}{M}\sum_t \mathcal{H}[\pi_\theta](s_t)$                  $\triangleright$ Compute entropy loss
24:             $L \leftarrow L_{\text{policy}} + c_1 L_{\text{value}} + c_2 L_{\text{entropy}}$     $(c_1 = 0.5, c_2 = 0.01)$           $\triangleright$ Total loss

25:             $\nabla_\theta L \leftarrow \frac{\nabla_\theta L}{\|\nabla_\theta L\|} \times \min\left(\|\nabla_\theta L\|, \text{max grad norm} = 0.5\right)$       $\triangleright$ Clip gradient norm
26:             $\theta \leftarrow \theta - \alpha\nabla_\theta L$     $(\alpha = 0.003)$                            $\triangleright$ Gradient step

27:             $\widehat{\text{KL}} \leftarrow \mathbb{E}_t\left[\tilde{r}_t(\theta) - 1 - \log\tilde{r}_t(\theta)\right]$                  $\triangleright$ Estimate KL divergence
28:             **if** $\widehat{\text{KL}} > 1.5 \times (\text{target KL} = \infty)$ **then break**           $\triangleright$ KL early stopping (disabled)
29:             **end if**
30:         **end for**
31:     **end for**
32: **end for**

---

