# OpenReview forum: "Counting atoms faster: policy-based nuclear magnetic resonance pulse sequencing for atomic abundance measurement"
_ICML.cc/2025/Conference — ICML 2025 poster_

### Official Review · Reviewer_PG82 · 2025-02-25

**Overall Recommendation:** 3

**Summary:**

The paper uses reinforcement learning (Proximal Policy Optimization algorithm) to learn policies to modulate NMR pulses for rapid atomic abundance quantification. The authors developed three interacting agents to (1) align nuclear spins for measurement, (2) facilitate rapid relaxation to equilibrium, and (3) coordinate control between these processes to reduce overall measurement time. Experiments were conducted in a simulated NMR environment using low-magnetic-field carbon-13 quantification for cost-effective, portable analysis of foodstuffs and soils. The results indicate notable performance improvements compared to conventional NMR pulse sequences, and the study also discusses the limitations of this technique.

## Update after rebuttal

While the rebuttal has substantially improved my understanding and appreciation of the contributions, I think that the practical issues that are not addressed by the simulator, i.e. coupling, T2* effects and shielded vs deshielded protons, make the current approach less convincing to be used as a replacement for physical NMR machines. However, if used as a POC or to provide an estimate then it is reasonable. I look forward for future improvements where the most common practical issues are well captured by the simulator. I have increased my score by 1 point to Weak Accept.

**Claims And Evidence:**

The claims in the paper are sound but not entirely convincing to me. The proposed approaches are theoretically sound. However, if I understand correctly, experiments were conducted mostly in a simulated environment of NMR. While in this setting the paper has demonstrated significant gains in terms of speed, my main problem is that I am not entirely sure how well the approaches work in practice. Practical issues like coupling (acknowledged in the paper), T2* effects and shielded vs de-shielded protons are not captured by the simulator presented here.

**Essential References Not Discussed:**

N/A

**Experimental Designs Or Analyses:**

I have a few questions regarding the experiments, in addition to the above comments:
1. On lines 170-173: How are the weights used in the simulator? Could you elaborate the term "class of spin"?
2. In section 3.2, does spoiling actually reset to the previous magnetization state for the sample even when net magnetization is 0? Let's say at 0 net magnetization, the number of protons whose direction is parallel to B0 is the same as the number of protons whose direction is anti-parallel to B0, would the spoiling agent create any zones where the protons are of the same polarity?
3. How do tuples $(\gamma, T_1, T_2)$ collected in section 2.2 remain valid for use in section 3.2 and 3.3 given that in 3.2 and 3.3 you have significantly cut down the relaxation time? Is there any guarantee that in practice this short amount of time is sufficient to achieve equilibrium after a period of high magnetization?

**Methods And Evaluation Criteria:**

The proposed methods and the evaluation criteria make sense to me. And indeed they are interesting. My only concern is that the paper seems to be incomplete as results mostly come from the simulated environment and there seems to be too little practical results other than the comparison of the chirping agents (sec 3.1) against collected 1D NMR pulses in section 2.2. However, these real data were used to train some of the agents.

I would like to see results when applying spoiling and toggled chirping and spoiling to real NMR pulse sequencing processes, not in a simulated environment. However such results are not presented here.

**Other Comments Or Suggestions:**

Could you give your opinion on modelling spin echo sequences (i.e. 180-degree pulses) in your simulator? Have you tried that?

**Other Strengths And Weaknesses:**

N/A

**Questions For Authors:**

N/A

**Relation To Broader Scientific Literature:**

If the proposed approaches work well in practice, it will be another example where reinforcement learning can substantially advance other fields.

**Theoretical Claims:**

The theoretical claims make sense to me. If backed up by experimental, non-simulated results this approach could be a game-changer in NMR pulse sequencing.

---

> ### Author Rebuttal · Authors · 2025-03-31
>
> We thank the reviewer for their comments and feedback. Our responses are:
>
> 1. **Regarding Claims and Methods.** The reviewer comments that the simulated approach is not satisfying. There is a precedent for the positive impact of simulated results in this field. Please refer to bullet point 4 in our response to Reviewer tJu3 as well as bullet point 1 in our response to Reviewer myr8.
> 2. **Regarding Experimental Design 1.** To obtain an FID in the simulator, the individual spins $(\gamma, \text{T1}, \text{T2})_i$ are separately simulated at first. The $(\gamma, \text{T1}, \text{T2})_i$ for fixed $i$ is what we refer to as a “class of spin.” Subsequently, we perform a weighted sum (over index $i$) of each of these magnetizations in order to get the observed $M_x$ and $M_y$. The weights in this sum are derived by a simple least squares fit to the empirical data and their relative variances were low enough that we did not feel them important to consider as sources of uncertainty.
> 3. **Regarding Experimental Design 2.** We agree that this is an important and subtle point. The agent does not have access to the net magnetization in the $B_0$ ($+z$) direction, and can only reasonably optimize to reduce transverse magnetization (in the $x$ and $y$ directions). The reviewer is correct that the state with $X$% of spins pointing in the $+z$ direction and $(1-X)$% of spins in the $-z$ direction parametrizes a line of maximum-reward states for the spoiling agent. If $X$ is nonzero, the spins in the $-z$ direction are unstable and precess back to $+z$, but this can cause longer-timescale oscillations in transverse magnetization than in the absence of active spoiling. This leads us to the rationale behind the second term in Equation (2) on lines 234-235: intermediate-timescale reductions in magnetization are insufficient, there must be a meaningful reduction in magnetization achieved by the end of the spoiling period.
> 4. **Regarding Experimental Design 3.** The description remains valid. Relaxation times T1 and T2 are defined as constant rates of decay of transverse magnetization in the presence of a strong background field $B_0$ and the absence of a perturbing field $B_1$ (analogous to half-life in radioactivity). By actively spoiling with an applied field $B_1$, we force a faster rate of decay, but this does not change the underlying model description. It only means we can not use an RL-based pulse to measure relaxation times T1 or T2, but it is not necessary that we be able to do so.
> 5. **Regarding Question 1** and spin-echo sequences. Our simulator can straightforwardly achieve a pi-pulse, this is just a matter of doubling the pulse time of the 90-degree pulse. It is intriguing to consider how introducing diverse pulse sequences into training (e.g. through teacher-forcing or by expanding exploration) could change the learned policy, especially given our response in Point 3. We are grateful for this suggestion, it is a very interesting avenue for future work and could unlock further performance benefits.

---

> > ### Comment · Reviewer_PG82 · 2025-04-04
> >
> > I thank the authors for their detailed rebuttal.While it has substantially improved my understanding and appreciation of the contributions, I still think that the practical issues that are not addressed by the simulator, i.e. coupling, T2* effects and shielded vs deshielded protons, make the current approach less convincing to be used as a replacement for physical NMR machines. However, if used as a POC or to provide an estimate then it is reasonable. I look forward for future improvements where the most common practical issues are well captured by the simulator.
> > I have increased my score by 1 point.

---

> > > ### Author Response · Authors · 2025-04-09
> > >
> > > We are grateful for the additional feedback and wish to affirm that the approach is not intended as a replacement for traditional NMR, and in fact its deployment in realistic environments will require cross-checking against such traditional NMR systems as outlined in L392-399 (right). We also wish to emphasize Point 2 in our initial response to Reviewer tJu3: while it is true that we did not capture all of these confounding variables in the simulation, in future work we will test both (A) improving the simulations and (B) developing a model-free paradigm using a hardware instantiation of the device wherein we train on pre-characterized samples. The feasibility of such a training paradigm in (B) is unique to this approach and will capture such effects when they are present in the analyte of interest. We are thankful that the Reviewer has raised these concerns, as they make clear we must select a sufficiently complex suite of analytes for the demonstration of (B).

---

### Official Review · Reviewer_tJu3 · 2025-03-12

**Overall Recommendation:** 4

**Summary:**

The paper presents another novel approach to performing NMR studies using reinforcement learning to modulate pulses. The authors study three different formulations: standard control of NMR pulse for spin alignment, relaxation of spins towards equilibrium, and allowing an agent to determine when relaxation (spoiling) should be allowed.

The trained agents could achieve the desired task efficiently in the experiments performed on simulated data for Carbon-13. This work generally represents a positive movement towards intelligent control of expensive procedures.

## Update after rebuttal

I was happy with the responses from the authors and feel the paper will make a great addition to the conference.

**Claims And Evidence:**

As far as the limitations of the evidence are clearly discussed, yes the claims are supported.

**Essential References Not Discussed:**

I did not notice important missing work.

**Experimental Designs Or Analyses:**

The experiments are designed well as a proof of concept.

**Methods And Evaluation Criteria:**

They make sense to the problem as a starting point from which more complex problems could be tackled.

**Other Comments Or Suggestions:**

The paper is well structured, and there were no apparent problems with the writing or figures.

**Other Strengths And Weaknesses:**

The idea is very interesting and well-formulated. Further, the decomposition of the problem into three stages is also well done. As the authors themselves mention, their data is limited not only in their use of simulation but also in that they study only a single species. Including real-world deficiencies of equipment and measurements would be a nice touch.

**Questions For Authors:**

1. Perhaps the authors can explain the main benefits of the RL approach over a non-intelligent control. Are there other semi-automatic approaches to controlling the chirp signals that require no training but are still efficient, something from MPC, for example?
2. What architectures were used in the studies, and how might this impact the results? Can time be introduced explicitly?
3. Can the state descriptions and other aspects of the RL training be extended to include multi-atom effects?

**Relation To Broader Scientific Literature:**

The use of agents to control and improve specific procedures is an emerging topic that will undoubtedly be of great interest to the community. From this perspective, the authors have made a positive contribution.

**Theoretical Claims:**

There are no large theoretical claims made in the study.

---

> ### Author Rebuttal · Authors · 2025-03-31
>
> We thank the reviewer for their comments and feedback. Our responses are:
>
> 1. **Regarding Weaknesses** and the inclusion of additional real-world equipment deficiencies. We agree that the inclusion of thermal instrument noise is only part of the picture. A variety of limitations are worth modeling, including spin participation (how many spins experience the influence of the strong background $B_0$ field) and magnetic field inhomogeneities (due to e.g. a low-quality magnet) [1], as well as spatial perturbations of the sample during measurement. Our perspective is that a hardware implementation of the apparatus will be necessary in order to meaningfully generate profiles for these features in a Bloch simulator, but that producing a dataset of these effects as generated by real hardware is both feasible and a promising avenue of future research.
> 2. **Regarding Question 1.** The main advantage of our approach over existing alternatives is the possibility of model-free development. We used a simulator as a proof-of-concept in this work to justify the construction of a bespoke hardware instantiation of the device and study its design parameters to reduce cost, but in a physical realization of the device we could simply learn a pulse by reinforcement on pre-characterized physical samples. Other approaches (including traditional NMR) require either analytical control or simulation of the underlying physics. In the Related Work, GRAPE is a result from optimal control, so is effectively the closest connection to MPC. However, GRAPE involves regression on a faithful model of the underlying system for a sample, which requires prior characterization and theoretical analysis of a compound of interest, and as such is only suited for sufficiently simple atomic or molecular systems that this characterization step can be performed. We have added a sentence explaining the connection between GRAPE and MPC in the Related Work and we welcome suggestions from the reviewer for relevant references to include which would further substantiate this connection.
> 3. **Regarding Question 2.** As the focus of this work was on an initial demonstration of the application with comparison to an analytically-derived baseline, and PPO suffices for this purpose, we did not attempt to further optimize performance by introducing other approaches such as TRPO, SAC, or A2C/A3C. Per bullet point 3 in our response to Reviewer myr8, we have included an Appendix explaining PPO and its alternatives. The timesteps referred to in the paper are easily relatable to explicit time within the simulation. We measure T2 and T1 in real-world seconds and compare timesteps against these values where appropriate (e.g. Figure 2), but we did not do so everywhere because timesteps are the temporal unit most suitable to analyzing the optimization process.
> 4. **Regarding Question 3.** Yes, techniques exist to extend Bloch simulations to account for a diverse range of molecular structure effects and it has been found that these are effective at improving MRI for molecularly complex samples such as brain tissue, e.g. [2]. Modification of the state descriptions and reward functions may be of interest for many reasons, an example is simultaneous tracking of mild impurities which one might reasonably treat as nuisance variables for the purpose of suppressing their radio-frequency response. In addition, whenever an analogous procedure exists in traditional physics-driven NMR to handle a specific such effect, we can optimize with respect to it in a similar fashion to the experiment in Section 3.1.2. Finally, we refer back to bullet point 2 above: wherever simulation-based methods fail, this method can in principle be applied directly on analytes of interest without an underlying physical representation of the system. This is not true of alternative approaches, though it comes with the caveats specified in Section 4 regarding distribution shift (lines 392-399, right side).
>
> [1] Ji, Yang, et al. "Dynamic B0 field shimming for improving pseudo‐continuous arterial spin labeling at 7 T." Magnetic Resonance in Medicine 93.4 (2025): 1674-1689.
>
> [2] Singh, Munendra, et al. "Bloch simulator–driven deep recurrent neural network for magnetization transfer contrast MR fingerprinting and CEST imaging." Magnetic resonance in medicine 90.4 (2023): 1518-1536.

---

> > ### Comment · Reviewer_tJu3 · 2025-04-02
> >
> > I thank the authors for their detailed responses. I would like to clarify, when asking about architecture, I was referring architecture of the actor. Are the authors using dense networks or something that can incorporate time like an LSTM / Transformers and so on. But to clarify, the authors write they use PPO but not A2C, so this is then PPO without baseline? Why not go something like A2C? The increase in complexity is minimal but the payoff can be significant.
> >
> > In any case, if the RL method (e.g. PPO, A2C and so on) along with the network architectures are included in the manuscript. I think the contribution is of interest to the community. Clearly real-world experiments would be of interest, but that is a big step up from simulation and would constitute research in an of itself. I will raise my score to an accept.

---

> > > ### Author Response · Authors · 2025-04-09
> > >
> > > We are grateful for the additional feedback and agree that it would be of future interest to produce results with a variety of treatments for the actor, to compare their learned behaviors as well as their relationships to comparable approaches in optimal control, e.g. the one we describe in our response to Reviewer MkXf in Point 2. We explain the stable-baselines3 implementation of PPO in detail in the new Appendix B, alongside alternative approaches which e.g. incorporate recurrent architectures (this implementation of PPO does not, it just uses a dense policy network but can be adapted to use transformers or LSTMs). In this paper we focused on 1D NMR as the key baseline, studied improvements relative to that baseline, and focused on developing a sensible methodology, since any such policy-based approach will raise similar questions to the ones motivating each experiment.

---

### Official Review · Reviewer_MkXf · 2025-03-17

**Overall Recommendation:** 3

**Summary:**

Counting atoms matters because it allows us to monitor and control the elemental makeup of materials, which is important for environmental sustainability applications. This paper presents a method that uses RL to optimise nuclear magnetic resonance (NMR) pulse sequences for faster and cheaper atomic abundance measurements.

## Update after rebuttal

I have increased my score by one point.

**Claims And Evidence:**

There are 3 main claims (contributions) in the paper:
1. "Train policies by reinforcement which shape and sequence magnetic field pulses for use in a simplified (low-field) NMR spectroscope". The method is claimed to be non-destructive and and generalisable to nuclear isotopes of many different elements.
2. "present a fast, robust simulator for generating large quantities of NMR spectroscopy data which is capable of reproducing the nuclear spin dynamics of many differ- ent samples in parallel when they are manipulated by an arbitrary magnetic field"
3. "A novel method to manipulate these spins for atomic abundance measurement, by training three inter-operating agents to orchestrate an NMR pulse sequence."

I think the experimental results back these claims in the sense that I can see there is an increased efficiency from metrics and graphs (e.g. model 2 reports 28% reduction in magnetisation, which I guess is good). However, as a non-expert with little to none knowledge in physics I can't assess how meaningful the experimental setup is and how significant the results are.

**Essential References Not Discussed:**

I don't know this field.

**Experimental Designs Or Analyses:**

I am not able to assess this.

**Methods And Evaluation Criteria:**

To my understanding, the simulator (based on Bloch equations) is calibrated using some real data (as described in Section 2.2). I have no idea if this is standard or if it makes sense for the things they are trying to demonstrate.

There aren't any baseline methods to compare the RL based approach to. Ideally, I'd like to see GRAPE (which this work is similar to), and potentially simpler baselines. E.g. would a random policy make sense here? Are there any "heuristic" policies that the field has developed previously?

The RL approach to policy learning itself is very standard - the authors use PPO.

**Other Comments Or Suggestions:**

NA

**Other Strengths And Weaknesses:**

On the ML side of things, the major weakness is lack of baselines.

As a non-expert I find it difficult to assess strengths and weaknesses more broadly.

**Questions For Authors:**

you mention GRAPE as a prior work related to what you do. Why is it not included as a baseline? More generally, what simpler baseline methods are there?

My review score is primarily due to the lack of baselines. I am not able to assess any of the physics, simulator setup etc.

**Relation To Broader Scientific Literature:**

I cannot assess the impact to the broader scientific literature.

**Theoretical Claims:**

there are no theoretical proofs in this work.

---

> ### Author Rebuttal · Authors · 2025-03-31
>
> We thank the reviewer for their comments and feedback. Our responses are:
>
> 1. **Regarding Weakness 1:** lack of a baseline. In fact, we do use the standard 1D NMR pulse as a baseline, and we compare against it in all experiments. The baseline we use is the simplest and most common approach to analyze atomic abundances of solution-phase samples in an NMR. While more sophisticated pulse sequences are often performed in the case of chemical characterization (which investigates e.g. which molecular bonds are present and what their relative positions are), relative abundance quantification is typically done by (A) performing a 90-degree pulse (~0.5 ms), (B) waiting around a second (a few T2s) for the system to equilbrate, (C) taking a Fourier transform of the FID, and (D) integrating the NMR spectrum. However, as stated on line 194, the chirp in Experiment 1 achieves its maximum magnetization state in 16% of the time required for this 1D NMR pulse in step (A). This is a 6x speedup even before accounting for step (B), which conservatively takes 100x longer than step (A), if not longer [1]. The black line in Figure 1 demonstrates that no analogous shortcut is possible with the 1D NMR baseline, i.e. one would have to perform (A)-(D) to obtain an equivalent result. Similar acquisition time reductions have been achieved in data-driven MRI [2].
> 2. **Regarding Question 1** and GRAPE [3] as a possible baseline. GRAPE requires specification of a target state. We are unaware of any standard state reported in the literature for atomic abundance measurement, but Footnote 1 on line 98 (page 2) shows how to derive one. The result seems paradoxical because it suggests performing no pulse at all. This seems strange, but is a consequence of differing assumptions and regimes of relevance. A zero pulse would work in theory, but requires different hardware which rapidly inserts and removes the sample from the magnetic field, so is not fully comparable to this approach. One can think of this as rapidly taking independent measurements of “signal plus background” and “only background” so that one can separate signal from the noise with a simple subtraction. Even if a zero pulse was found to perform better in simulation, engineering such hardware is vastly more challenging than a stationary approach; a relevant point of comparison is “magic angle spinning” in solid-state NMR where small cuvettes of material are spun at MHz frequencies [4]. We also emphasize that, besides performance considerations, in our approach there is no fundamental requirement of a faithful simulation of the underlying system. In a hardware instantiation, we can simply learn by reinforcement on real pre-characterized samples, which is not the case for GRAPE. The underlying reason for these differences is that GRAPE assumes the quantum mechanical regime is an appropriate description of the spins (i.e. samples contain only a few atoms or molecules so that quantum effects dominate), whereas RL on a Bloch simulator assumes a semi-classical description (i.e. samples contain multiple Avogadro’s numbers of atoms or molecules whose nuclear spins together behave classically). Areas of application appropriate to GRAPE include e.g. quantum computing [5,6] where individual molecules are manipulated into specific quantum states and then controlled as components of logic gates. Areas of application appropriate to our method include fields which commonly process macroscopic (gram-scale) samples appropriate to everyday contexts, such as food science, agriculture, and human health. However, the reviewer’s comment suggests a nontrivial avenue of further investigation which we had not realized earlier, and we are grateful for this insight. The top left of Figure 3 shows that a preferred direction is learned by the policy (black dots), providing an alternative target state for a GRAPE analysis than that derived in Footnote 1. It would be interesting to understand whether GRAPE with this target state is equivalent to our learned policy. We have added a sentence to the Future Work section incorporating this observation.
>
> [1] Joseph, David, and Christian Griesinger. "Optimal control pulses for the 1.2-GHz (28.2-T) NMR spectrometers." Science Advances 9.45 (2023): eadj1133.
>
> [2] Ma, Dan, et al. Magnetic resonance fingerprinting. Nature, 495(7440):187–192, 2013.
>
> [3] Khaneja, Navin, et al. "Optimal control of coupled spin dynamics: design of NMR pulse sequences by gradient ascent algorithms." Journal of magnetic resonance 172.2 (2005): 296-305.
>
> [4] Reif, Bernd, et al. "Solid-state NMR spectroscopy." Nature Reviews Methods Primers 1.1 (2021): 2.
>
> [5] Jones, Jonathan A. "Controlling NMR spin systems for quantum computation." Progress in Nuclear Magnetic Resonance Spectroscopy 140 (2024): 49-85.
>
> [6] Schulte-Herbrüggen, T., et al. "Optimal control for generating quantum gates in open dissipative systems." Journal of Physics B: Atomic, Molecular and Optical Physics 44.15 (2011): 154013.

---

> > ### Comment · Reviewer_MkXf · 2025-04-07
> >
> > [Apologies, I didn't realise I had to post my comment as "rebuttal comment", so I'm posting it again]
> >
> > Thanks for your thorough response.
> >
> > Overall, I think the "Application driven ML" is a great idea, and this paper presents an interesting application (though I apologise I do not have the sufficient background to appreciate it). I'll increase my score by one point.

---

> > > ### Author Response · Authors · 2025-04-09
> > >
> > > We are grateful for the reviewer’s time and constructive responses.

---

### Official Review · Reviewer_myr8 · 2025-03-24

**Overall Recommendation:** 4

**Summary:**

The paper applies reinforcement learning to NMR data, to infer the elemental composition of a sample. They learn optimal policies to modulate NMR pulses for elemental abundance quantification. They present promising results on simulated data.

## update after rebuttal

I have raised my score.

**Claims And Evidence:**

Yes

**Essential References Not Discussed:**

I am not sufficiently familiar with the literature to comment.

**Experimental Designs Or Analyses:**

Yes

**Methods And Evaluation Criteria:**

Yes

**Other Comments Or Suggestions:**

I did not find any typos.

**Other Strengths And Weaknesses:**

The paper only presents application to simulated NMR data. It would be nice to have real data analyzed.

The machine learning content of the paper is weak, and it would seem that this paper is better suited to a specialized journal focusing on NMR.

**Questions For Authors:**

I understand the authors are using algorithms developed elsewhere. But given the readership of ICML I think readers would appreciate some more details were included about the algorithm used (things like PPO, MDP, etc.). Can the authors include a summarized self-contained description of these methods?

**Relation To Broader Scientific Literature:**

I am not sufficiently familiar with the literature to comment.

**Theoretical Claims:**

Yes

---

> ### Author Rebuttal · Authors · 2025-03-30
>
> We thank the reviewer for their comments and feedback. Our responses are:
>
> 1. **Regarding Weakness 1.** The reviewer states that the paper only presents application to simulated data. This is not totally accurate, for 2 reasons. **(I)** Firstly, in Experiment 1, the spin sets used in both training and evaluation are derived from genuine raw NMR data (a serial dilution of caffeine in water with a 90-degree pulse), and the Bloch model is a faithful representation of the underlying system. This is partly evidenced by the fact that, using the Bloch model, we can reproduce the empirically observed spectra exactly. **(II)** Secondly, we did investigate how to operate real-world NMR hardware with a control policy, but found it is not practical with current machines such as those used to produce the empirical data, as they are only operable with proprietary software, and purchasing a dedicated setup for modification would cost in the 6- to 7-figure range (USD). Such investment of funds and effort is challenging to justify without a proof-of-concept, which our results do provide. Furthermore, our results demonstrate concrete evidence that a high-fidelity, laboratory-grade NMR with expensive magnets, liquid nitrogen cooling, and similar luxury features can be potentially replaced with lower-quality, less expensive benchtop hardware. As a consequence of investigating this proof-of-concept, not only did we learn that the method has theoretical promise, but we also obtained evidence that the magnitude of investment required to explore hardware development could be much lower than this initial expectation. This finding is not obvious, and it represents an important motivating result which will be a precursor to a longer-term hardware development effort.
>
> 2. **Regarding Weakness 2.** The reviewer comments that the machine learning content is weak, and that the paper is better suited to a venue focused on NMR. The ICML 2025 Call for Papers states the following regarding the Application-Driven Machine Learning track: “Application-Driven Machine Learning (innovative techniques, problems, and datasets that are of interest to the machine learning community and driven by the needs of end-users in applications such as healthcare, physical sciences, biosciences, social sciences, sustainability and climate, etc.)” As noted in the Related Work section, there is little existing work applying machine learning specifically to NMR pulse sequencing and atomic abundance measurement, so this work falls under the category of a “problem of interest” in the language of the Call. NMR is a common measurement technique applied within healthcare, analytical chemistry, and bioscience and is the basis of MRI technology. (The application specifically to sustainability is somewhat novel, though sustainability research has significant overlap with the aforementioned 3 fields.) While we agree that multiple field-specific venues could be acceptable for such a work, the key point we are trying to demonstrate and emphasize with this paper is that an application of machine learning unlocks capabilities which can impact many different areas of science in a unified fashion. In conjunction with the language of the Call, this widespread impact thanks to open-source frameworks and developments in reinforcement learning (including the Gymnasium framework and the PPO family of policy gradient methods used to produce our results) was our primary reasoning to submit to ICML 2025.
> 3. **Regarding Question 1.** Given the diverse scientific backgrounds of potential readers for this work, we agree that it is appropriate to include additional definitions regarding MDPs, PPO-based agents, the designed objective functions, and potential alternatives. We have added an Appendix B where we provide these, and cite relevant related works.

---

> > ### Comment · Reviewer_myr8 · 2025-04-04
> >
> > I thank the authors for their response. I have increased my score.

---

> > > ### Author Response · Authors · 2025-04-09
> > >
> > > We are grateful for the reviewer’s time and constructive responses.

---

### Decision · Program_Chairs · 2025-05-01

**Decision:**

Accept (poster)

**Comment:**

**Summary.**

The topic of this work is to determine the elemental composition of a material using Nuclear Magnetic Resonance (henceforth NMR).

Specifically, the authors propose a reinforcement-learning based method (Proximal Policy Optimization) to modulate the NMR pulses for quantification of the element composition of a sample.

Claims: their method is faster, non-destructive, more versatile and less costly than traditional ones.

They further demonstrate the applicability of their method through experiments.

**Important note.**

The authors mentioned that their paper should be assessed as an “application-driven” paper (special track).

**Strengths.**

* The method is superior to non-ML existing alternatives because it opens the possibility of model-free development. This is an important point for the “application-driven” paper track since “prior work outside the ML literature should be considered”.

**Weaknesses.**

* The ML content of the paper was deemed to be light (myr8). However, since this is an “Application-Driven ML Submission”, introducing a “problem of interest” to the ICML community can be a sufficient contribution.
* Understanding the paper is challenging for the non-domain-expert (MkXf).
* It was raised by PG82 that “coupling, T2* effects and shielded vs deshielded protons, make the current approach less convincing to be used as a replacement for physical NMR machines”. This demonstrates a weakness compared to alternative methods.

**Discussion and reviewer consensus.**

The author response was particularly effective at dispelling most of the reviewers’ concerns; everyone has increased their score, and there is a consensus that the paper could be accepted.

**Additional remarks.**

One concern was the lack of baselines in the experiments (MkXf); however, this concern was addressed: the authors use the standard 1D NMR pulse as a baseline in all their experiments, and argued that using so-called GRAPE was not appropriate here.

**Overall evaluation.**

If there is room, I think the paper would be a nice addition to the program.